

# Auto-correlation of journal impact factor for consensus research reporting statements: a cohort study

Daniel R. Shanahan

BioMed Central, London, United Kingdom

## ABSTRACT

**Background.** The Journal Citation Reports journal impact factors (JIFs) are widely used to rank and evaluate journals, standing as a proxy for the relative importance of a journal within its field. However, numerous criticisms have been made of use of a JIF to evaluate importance. This problem is exacerbated when the use of JIFs is extended to evaluate not only the journals, but the papers therein. The purpose of this study was therefore to investigate the relationship between the number of citations and journal IF for identical articles published simultaneously in multiple journals.

**Methods.** Eligible articles were consensus research reporting statements listed on the EQUATOR Network website that were published simultaneously in three or more journals. The correlation between the citation count for each article and the median journal JIF over the published period, and between the citation count and number of article accesses was calculated for each reporting statement.

**Results.** Nine research reporting statements were included in this analysis, representing 85 articles published across 58 journals in biomedicine. The number of citations was strongly correlated to the JIF for six of the nine reporting guidelines, with moderate correlation shown for the remaining three guidelines (median $r = 0.66$, 95% CI [0.45–0.90]). There was also a strong positive correlation between the number of citations and the number of article accesses (median $r = 0.71$, 95% CI [0.5–0.8]), although the number of data points for this analysis were limited. When adjusted for the individual reporting guidelines, each logarithm unit of JIF predicted a median increase of 0.8 logarithm units of citation counts (95% CI [−0.4–5.2]), and each logarithm unit of article accesses predicted a median increase of 0.1 logarithm units of citation counts (95% CI [−0.9–1.4]). This model explained 26% of the variance in citations (median adjusted $r^2 = 0.26$, range 0.18–1.0).

**Conclusion.** The impact factor of the journal in which a reporting statement was published was shown to influence the number of citations that statement will gather over time. Similarly, the number of article accesses also influenced the number of citations, although to a lesser extent than the impact factor. This demonstrates that citation counts are not purely a reflection of scientific merit and the impact factor is, in fact, auto-correlated.

Corresponding author
Daniel R. Shanahan,
daniel.shanahan@biomedcentral.com

## INTRODUCTION

The journal impact factor (JIF) was introduced as a method to compare journals regardless of the number of articles they publish, to help select journals for the Science Citation Index (SCI), recognizing that small but important journals would not be chosen based solely on absolute publication or citation counts (*Brodman, 1960*).

Today, JIFs are widely used to rank and evaluate journals, as a proxy for the relative importance of a journal within its field; journals with higher JIFs are in general deemed to be more important than those with lower ones (*Brembs, Button & Munafò, 2013*; *Eyre-Walker & Stoletzki, 2013*). This application of the JIF is also widely extended to the evaluation of individual scientists, with the JIF of the journals they have published in used as a precondition for grants or faculty promotion and tenure (*Archambault & Larivière, 2009*; *Fuyono & Cyranoski, 2006*).

There have been many criticisms levelled at the use of JIF to evaluate a journal's importance. It has been demonstrated that JIFs, and citation analysis in general, are affected by field-dependent factors (*Archambault & Larivière, 2009*; *Bornmann & Daniel, 2008*), which may invalidate comparisons not only across disciplines but even within different fields of research of one discipline (*Anauati, Galiani & Gálvez, 2014*). Furthermore, the JIF is based on the arithmetic mean number of citations per paper, yet it is obvious that citation distribution is not normal, but follows a power-law distribution—20% of papers in a journal can account for 80% of the total citations—making it a statistically inappropriate measure (*Archambault & Larivière, 2009*; *Pulverer, 2015*; *Adler, Ewing & Taylor, 2009*). This means that a small number of highly-cited articles contribute disproportionately to a journal's JIF, with most articles within a journal receiving fewer citations than would be predicted from the JIF (*Anonymous, 2005*).

These problems are exacerbated when the use of JIFs is extended to evaluate not only the journals, but papers therein (*Eyre-Walker & Stoletzki, 2013*; *Pulverer, 2015*). A number of studies have shown that articles published in journals with high JIFs are cited more frequently than those published in lower-JIF journals (*Callaham, Wears & Weber, 2002*; *Seglen, 1994*; *Nieminen et al., 2006*; *Filion & Pless, 2008*; *Etter & Stapleton, 2009*), with the implication that research articles published in high-JIF journals have a greater impact on the scientific literature and are perceived to be of better quality than those published in less prominent journals.

However, the number of citations a paper receives has been found to be an extremely error-prone measure of scientific merit, with poor correlation observed between article quality as rated by external assessors and the number of citations the article receives (*Eyre-Walker & Stoletzki, 2013*). Furthermore, a number of external factors have been shown to influence the citation rate of articles (*Bornmann & Daniel, 2008*), including the publication licence (open access vs. subscription) (*Piwowar & Vision, 2013*), number of authors, contributing institutions (*Figg et al., 2006*; *Stringer, Sales-Pardo & Nunes Amaral, 2010*) and the number of article accesses (*Watson, 2009*; *Schloegl & Gorraiz, 2010*; *Paiva, Lima & Paiva, 2012*).

Due to these influences, evaluating whether the JIF is correlated with citations counts requires identical articles published simultaneously across multiple journals. *Larivière & Gringras (2010)* compared the number of citations to 4,532 pairs of duplicate publications (based on identical titles, first author and length of reference list), showing that the citation impact of duplicates published in the journals with a higher JIF was almost double that of those published in lower-JIF journals. This provided evidence for preferential attachment—also known as the Matthew effect (*Merton, 1968*)—demonstrating a cumulative advantage for high-JIF journals, whereby 'the rich get richer and the poor get poorer.' However, this comparison was limited to pairs of publications, which prevented a more nuanced evaluation of the link between the journal JIF and number of citations the articles received.

Unlike research papers, reporting guidelines are often published simultaneously to encourage wider adoption and dissemination, as they are not scientific publications, but rather recommendations for authors. *Perneger (2010)* investigated the presence of journal-bias using the reporting guidelines QUOROM, CONSORT, STARD and STROBE, finding a strong correlation between the JIF and the number of citations to each publication.

The advantage of investigating the Matthew effect for journals using reporting guidelines is that they are published in multiple journals and control for many of the factors that influence citation counts. However, Perneger's study did not take into account the number of article accesses, which has also been shown to be correlated with the number of citations (*Watson, 2009*; *Schloegl & Gorraiz, 2010*; *Paiva, Lima & Paiva, 2012*). Furthermore, since the publication of Perneger's study, the number of reporting guidelines has increased dramatically providing a larger base for comparison.

The purpose of this study was to investigate the correlation between JIF and the number of citations to consensus research reporting statements published simultaneously in multiple journals, and the correlations between the total article accesses according to the COUNTER code of practice (*Pesch, 2015*) and number of citations. The hypothesis was that there will be an observable Matthew effect for consensus reporting statements published simultaneously across multiple journals, with articles in high-JIF journals being more highly-cited than those in lower-JIF journals, independent of the number of article accesses that statement received.

## METHODS

Eligible articles were consensus statements on research reporting listed on the EQUATOR Network website (http://www.equator-network.org/) that were published simultaneously in identical formats in three or more journals, each with different citations details. Republished articles—those which cited another version of the guideline, stated that they were republished within the article or not published simultaneously (or in the next available issue)—were excluded. Research reporting statements that directly cited another version of the reporting guideline as definitive were excluded, as were those published after 31 December 2013, because the citation data were not yet fully available at the time of data extraction.

The JIFs were obtained from the Journal Citation Reports (JCR), with a one-year offset (e.g., using the 2013 JIF value for articles in 2014), as this would have been the JIF attributed

to the journal during the citation accumulation period. Journals that were not indexed in the JCR were included in this analysis for completeness as they would also accumulate citations and article accesses, which would impact on those received by the other journals, but were assigned an JIF of zero, to represent their position at the bottom of the relevant JCR category.

Total citations counts were obtained for all these articles using the abstract and citation database Scopus (Elsevier B.V., http://www.scopus.com), on the same day (22/07/2015); this allowed coverage of journals without an JIF, but restricted the citation counts to those accumulated within the scientific literature.

Total article access counts according to the COUNTER code of practice were obtained between 20 July 2015 and 24 July 2015 from the publishers or the journal website, where available. In many cases, these data were not available either due to errors, system migrations or the publishers' not collecting the data according to the COUNTER standards.

## Statistical analyses

Data were analysed using Microsoft Excel 2010. Analyses were conducted for each reporting guideline separately. Spearman's correlation coefficient was calculated for correlations between the total number of citations to the each article and the JIF, and the total number of citations and the number of article accesses. As the JIF follows a Bradford (power-law) distribution (*Black, 2004*), the median value for each journal over the published period of the article was used, assigning a value of zero for years when the journals were not included in the JCR.

The number of citations predicted by the JIF and the number of article accesses were also calculated in a general linear model, stratified by reporting statement. As the distributions of citations, JIFs and article accesses are skewed towards high-values, these variables were transformed using the logarithm in base 10 for this calculation (excluding null values). For each statement and comparison, Pearson's correlation coefficient was also calculated for the transformed values.

Finally, multivariate regression analysis was performed, considering the combination of both JIF and total article accesses as a predictor for citations.

## RESULTS

Overall, 280 reporting guidelines were evaluated for inclusion in this analysis; of these, 13 were associated with three or more publications. Four guidelines were excluded: one as it was published too recently for there to be robust citation data (TRIPOD), and three as the articles were republications, citing a single definitive version (QUOROM, ARRIVE and SQUIRE). Therefore, nine reporting guidelines were included in this analysis, representing 85 articles published across 58 journals in biomedicine. The journals included in this analysis are included in the Appendix.

Table 1 shows basic information about the nine reporting guidelines included. The median number of published articles per reporting guideline was 10 (range 5–15), published between 2003 and 2013. Articles that were not published simultaneously (defined as same publication date for continuous publication journals, or next available issue) were excluded from the analysis. The range of the journal 2013 JIFs was 0–39.4.

**Table 1** Descriptive statistics for consensus research reporting statements included in the analyses.

| Reporting guideline | Year of publication | No. of journals | Median 2013 IF (range) | Median no. of citations (range) | Median no. of accesses (range) |
|---|---|---|---|---|---|
| STARD | 2003 | 15 | 2.23 (1.37–16.4) | 39 (3–781) | 1,257 (615–12,203) |
| STROBE | 2007 | 10 | 6.18 (0.600–39.2) | 150 (27–966) | 10,057 (142–17,055) |
| PRISMA | 2009 | 7 | 5.48 (0–16.4) | 962 (3–2,941) | 10,696 (7,644–49,961) |
| STREGA* | 2009 | 8 | 5.15 (0–16.1) | 50 (14–109) | 2,233 (1,977–3,873) |
| CONSORT | 2010 | 12 | 4.37 (0–16.4) | 134 (2–917) | 6,591 (2,582–13,027) |
| REFLECT** | 2010 | 5 | 2.07 (0.771–2.51) | 12 (3–30) | 1,522 (856–3,394) |
| GRIP | 2011 | 11 | 5.34 (0–16.4) | 5 (0–25) | 1,331 (159–1,823) |
| CARE | 2013 | 7 | 0 (0–5.48) | 5 (0–20) | 1,326 (137–14,886) |
| CHEERS | 2013 | 10 | 2.89 (0–16.4) | 14 (4–42) | 951 (303–6,091) |
| **Overall** | | **85** | **3.01 (0–39.2)** | **27 (0–2,941)** | **2,233 (137–49,961)** |

**Notes.**
  *STREGA is an extension of the STARD guidelines.
  **REFLECT is an extension of the CONSORT guidelines.

The number of citations to individual articles ranged between 0 and 2,941. The median number of citations varied depending on the guideline in question, and ranged between 5 and 962. The number of individual article accesses ranged between 137 and 49,961. The median number of articles accesses according to reporting guideline varied between 951 and 10,696.

Spearman's correlation analysis showed a median correlation of 0.66 between the JIFs of the source journal and the number of citations (95% CI [0.45–0.90]; Table 2), with six of nine analysed reporting guidelines showing statistical significance ($p < 0.05$). The four guidelines that did not reach statistical significance demonstrated a weaker correlation between JIF and the number of citations (0.45 for STREGA, 0.52 for GRIP, 0.61 for CARE and 0.33 for CHEERS). Analysis of logarithm-transformed values using Pearson's correlation coefficient revealed a median correlation of 0.83 between JIF and the number of citations (95%CI [0.43–0.81]; Table 2), with all but two reaching statistical significance (0.43 for STREGA and 0.81 for REFLECT).

These results were supported by linear regression analysis (Table 2), with doubling the JIF associated with a 3-fold increase in citations (1:1.2 logarithmic units, range 0.5–3.0; Table 2 and Fig. 1).

Peer J

**Table 2   Correlations between citations, and journal impact factor and article downloads, and regression coefficients for logarithm-transformed values.**

| | STARD | STROBE | PRISMA | STREGA | CONSORT | REFLECT | GRIPS | CARE | CHEERS |
|---|---|---|---|---|---|---|---|---|---|
| **Correlation between citations and journal IF** | | | | | | | | | |
| Spearman's correlation coefficient | 0.66 | 0.93 | 0.86 | 0.45 | 0.89 | 0.90 | 0.52 | 0.61 | 0.33 |
| | ($p = 0.008$) | ($p < 0.001$) | ($p = 0.014$) | ($p = 0.26$) | ($p < 0.001$) | ($p = 0.037$) | ($p = 0.099$) | ($p = 0.15$) | ($p = 0.36$) |
| Pearson's correlation coefficient (logarithms) | 0.86 | 0.99 | 0.83 | 0.43 | 0.86 | 0.81 | 0.43 | 0.93 | 0.63 |
| | ($p < 0.001$) | ($p < 0.001$) | ($p = 0.02$) | ($p = 0.3$) | ($p < 0.001$) | ($p = 0.1$) | ($p = 0.02$) | ($p = 0.003$) | ($p = 0.05$) |
| **Correlation between citations and article downloads** | | | | | | | | | |
| Spearman's correlation coefficient | 1.0 | 0.94 | 0.8 | 0.8 | 0.71 | 0.5 | 0.03 | 0.60 | 0.61 |
| | | ($p = 0.005$) | ($p = 0.2$) | ($p = 0.2$) | ($p = 0.11$) | ($p = 0.67$) | ($p = 0.96$) | ($p = 0.28$) | ($p = 0.08$) |
| Pearson's correlation coefficient (logarithms) | 1.0 | 0.78 | 0.6 | 0.79 | 0.19 | 0.35 | −0.01 | 0.58 | 0.63 |
| | ($p = 0.04$) | ($p = 0.07$) | ($p = 0.4$) | ($p = 0.1$) | ($p = 0.7$) | ($p = 0.8$) | | ($p = 0.3$) | ($p = 0.07$) |
| **Linear regression coefficients (95% confidence intervals)** | | | | | | | | | |
| Logarithm of citations per logarithm of IF | 1.3 | 0.9 | 1.6 | 0.3 | 1.2 | 1.6 | 0.5 | 3.0 | 0.6 |
| | (0.8–1.9) | (0.8–1.1) | (0.1–3.0) | (−0.5–1.1) | (0.6–1.8) | (−0.5–3.7) | (−0.6–1.6) | (−13.8–18.9) | (−0.1–1.4) |
| Logarithm of citations per logarithm of accesses | 1.1 | 0.6 | 2.1 | 2.6 | 0.4 | 0.2 | −0.01 | 0.2 | 0.4 |
| | (0.1–2.1) | (−0.1–1.3) | (−6.4–10.7) | (−3.4–8.6) | (−2.8–3.7) | (−7.6–8.1) | (−1.9–1.9) | (−0.4–0.9) | (−0.04–0.9) |
| **Multivariate regression coefficients (95% confidence intervals)** | | | | | | | | | |
| Logarithm of citations per logarithm of IF | -0.4 | 0.8 | 4.5 | −1.1 | 0.9 | 14.2 | 0.7 | 5.2 | 0.4 |
| | | (0.7–1.0) | (−20–29) | (−24–22) | (−0.7–2.5) | | (−3.7–5.2) | | (−0.8–1.5) |
| Logarithm of citations per logarithm of accesses | 1.4 | 0.1 | −4.3 | 5.5 | −0.3 | 0.6 | −0.1 | −0.9 | 0.3 |
| | | (0.0–0.2) | (−42–33) | (−60–71) | (−3.6–3.0) | | (−2.8–2.7) | | (−0.6–1.2) |

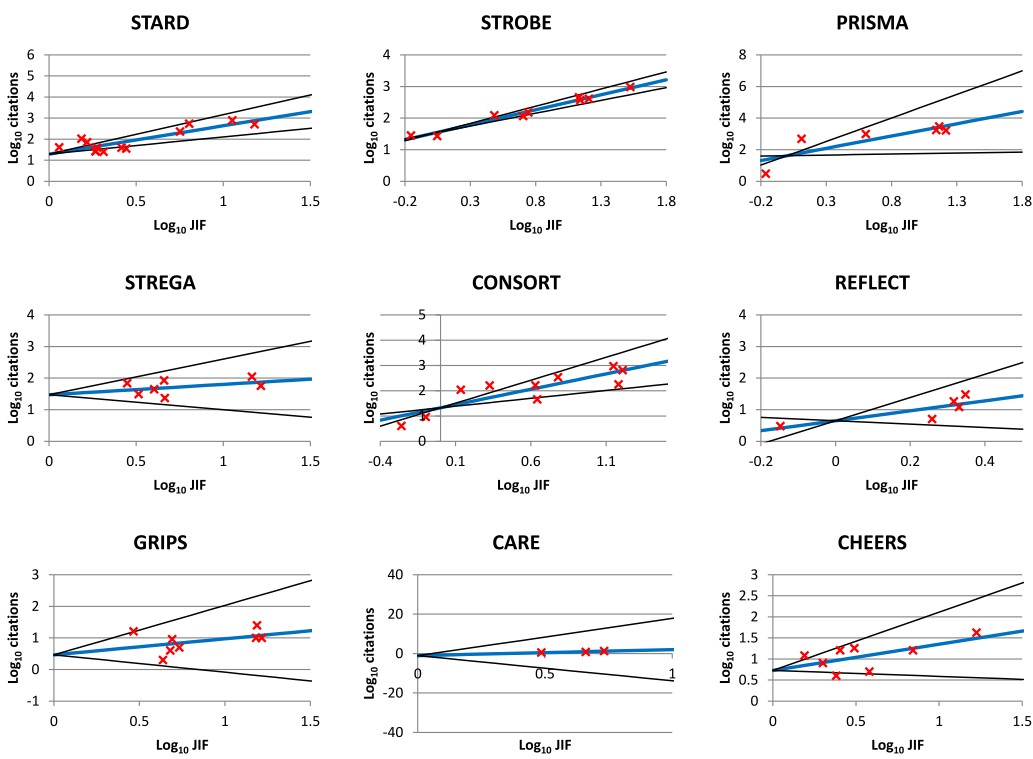

**Figure 1** **Linear regression fits for logarithm-transformed journal impact factor and citations for nine co-published consensus reporting statements.** Red crosses represent the raw data, the blue line the regression fit line and the black lines the 95% confidence intervals for the regression analysis.

A median correlation of 0.71 was observed between article accesses and citations, using Spearman's correlation coefficient (95% CI [0.5–0.8]; Table 2) and 0.6 using Pearson's coefficient for the logarithm-transformed values (95% CI [0.19–0.78]; Table 2), for all reporting guidelines except GRIPS, which showed weak-to-no correlation. Univariate linear regression showed that increasing the article accesses by a factor of 10 was associated with a 3.2-fold increase in citations counts (1:0.5 logarithmic units, range 0.3–0.6), with trend of article accesses having greater impact on citation counts for older reporting statements (Table 2).

I also conducted multivariate analysis at the reporting guideline level, considering both JIF and total article accesses as predictors for citations (Table 2). When adjusted for the individual reporting guidelines, increasing the JIF by a factor of 10 predicted a median 6.3-fold increase in citation counts (1:0.8 logarithmic units, 95% CI [−0.4–5.2]), while article accesses needed to increase by a factor of 50 to achieve the same increase in citation counts (1:0.1 logarithmic units, 95% CI [−0.9–1.4]). This model explained a median 26% of the variance in citations (median adjusted $r^2 = 0.26$, range 0.18–1.0), but only reach statistical significance for STROBE ($p = 0.0001$).

## DISCUSSION

This analysis extends the work by *Perneger (2010)*, to show that both the JIF of the journal in which an article is published, and the number of accesses that article receives are strongly correlated with the number of citations to that article. Identical articles published simultaneously in multiple journals were used for this analysis, which controlled for variations in the characteristics of the articles themselves (for example, scientific content, number of authors and institutions); therefore, any variation in the citations count for the article reflect either the journal they are published in or the number of article accesses.

The results demonstrated a strong correlation between the number of citations and the JIF and between the number of citations and the number of accesses for most of the reporting guidelines included, suggesting the presence of a Matthew effect. The regression slope for citations and article accesses demonstrated a weaker influence of article accesses on the total number of citations. This was also supported by the multivariate regression analysis, in which the coefficient for article accesses was lower than for JIF for every consensus reporting statement analysed, except PRISMA and STREGA. This suggests that JIF has a proportionally greater effect on the number of citations to an article.

The calculated correlation coefficients for STARD and STROBE were higher than those calculated by Perneger (0.66 vs 0.65 for STARD, and 0.93 vs 0.81 for STROBE, respectively) (2010). Similarly, the influence of article accesses on the number of citations generally increased with the time since publication. This could be explained by early citation counts influencing later and total article citations, as was previously demonstrated by *Adams (2005)*. However, article accesses and citations have a different obsolescence pattern (*Schloegl & Gorraiz, 2010*), with accesses rapidly accumulating on publication, while citations take longer to build. Due to these different obsolescence patterns, it is essential, to consider a time-window of several years, which could explain the increasing regression coefficients with the age of the publication.

A number of factors have been previously shown to influence the citation rate of articles (*Bornmann & Daniel, 2008*), including the publication licence (open access vs. subscription) (*Piwowar & Vision, 2013*), number of authors, contributing institutions (*Figg et al., 2006*; *Stringer, Sales-Pardo & Nunes Amaral, 2010*) and the number of article accesses (*Watson, 2009*; *Schloegl & Gorraiz, 2010*; *Paiva, Lima & Paiva, 2012*); however, by using consensus reporting statements, this study was able to control for many of these factors. *Lozano, Larivière & Gingras (2012)* have further suggested a weakening relationship between the JIF and article citations since 1990.

The results of the current study suggest that, rather than the JIF serving as a proxy for scientific value of the journal or article, it is instead auto-correlated, with articles being cited simply because they are published in a high-JIF journal. This may help to explain the stability of journal rankings within the JCR.

### Limitations

There are a number of limitations to this analysis, requiring a more cautious interpretation of the results. As JIF are associated with prestige and audience size, high-JIF journals are often able to dedicate more resources to promotional activity surrounding publications,

which could directly influence both the article accesses and citation counts; however, this was not accounted for in this study. Similarly, while translations were included if they were published simultaneously with the other statements, the effect of language on citation distribution was not investigated in the current study.

There are a limited number of data points included in the correlation calculations, which could lead to an overestimation of the correlation coefficients, particularly for analysis of article accesses. This was due to errors in the article counters, data being lost during system migrations or the publishers' not collecting that data. There is also the confounding issue of publishers collecting article access statistics in a non-COUNTER compliant manner, and the rapid increase in e-journal use between 2001 and 2006 (*Schloegl & Gorraiz, 2010*), which directly impacted on the article access statistics for reporting statements published before 2006 (STARD).

While this study restricted the citation data to the Scopus database, citation data might be obtained from different sources, including the Web of Science, Google Scholar, PubMed and others, with known inconsistencies between them (*Durieux & Gevenois, 2010*). Therefore, analyses using different sources for the total number of citations may observe different trends.

The analysis was also limited to biomedical consensus research reporting statements, rather than original research. As these articles are typically published simultaneously in multiple journals under an open access licence, I was able to control for many confounding variables. However, different article types are known to display different citation patterns (*Nieder, Dalhaug & Aandahl, 2013*) and, as such, the patterns seen here may not be applicable to other article types or different fields of research.

These guidelines were also published in journals from quite a heterogeneous selection of medical specialties. Field-dependent factors are known to affect citation analysis, even within disciplines (*Anauati, Galiani & Gálvez, 2014*); therefore, these fields may have distinct publication and citation behaviour, which would influence the generalizability of the results.

## CONCLUSION

In this study, the impact factor of the journal in which a consensus reporting statement was published was shown to correlate with the number of citations that statement will gather over time. Similarly, the number of article accesses also influenced the number of citations, although to a lesser extent than the impact factor. These findings suggest that citation counts are not purely a reflection of scientific merit, and the journal impact factor is, in fact, auto-correlated.

**Abbreviations**

| | |
|---|---|
| **ARRIVE** | Animals in Research: Reporting *In Vivo* Experiments |
| **CARE** | CAse REport guidelines |
| **CHEERS** | Consolidated Health Economic Evaluation Reporting Standards |
| **CONSORT** | CONsolidated Standards of Reporting Trials |
| **EQUATOR** | Enhancing the QUAlity and Transparency Of health Research |

| | |
|---|---|
| **GRIPS** | Genetic RIsk Prediction Studies |
| **JIF** | journal impact factor; |
| **JCR** | Thomson Reuters' Web of Science Journal Citation Reports |
| **PRISMA** | Preferred Reporting Items for Systematic Reviews and Meta-Analyses |
| **QUOROM** | Quality of Reporting of Meta-analyses |
| **REFLECT** | Reporting guidelines for randomized controlled trials in livestock and food safety |
| **SCI** | Science Citation Index |
| **SQUIRE** | Standards for QUality Improvement Reporting Excellence |
| **STARD** | Standards for Reporting of Diagnostic Accuracy |
| **STREGA** | STrengthening the REporting of Genetic Association Studies |
| **STROBE** | Strengthening the Reporting of Observational Studies in Epidemiology |
| **TRIPOD** | Transparent Reporting of a multivariable prediction model for Individual Prognosis Or Diagnosis |

## ACKNOWLEDGEMENTS

The author would like to thank David Moher for his advice and support with both data analysis and the first version of this article.

### Funding

The author received no funding for this work.

### Competing Interests

DRS is an employee of BioMed Central.

### Author Contributions

- Daniel R. Shanahan conceived and designed the experiments, analyzed the data, wrote the paper, prepared figures and/or tables, reviewed drafts of the paper.

### Data Availability

The data and analysis from this article are included in the Supplemental files.

### Supplemental Information

Supplemental information for this article can be found online at http://dx.doi.org/10.7717/peerj.1887#supplemental-information.

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
