# Peer review of "Auto-correlation of journal impact factor for consensus research reporting statements: a cohort study"

_PeerJ, doi:10.7717/peerj.1887_

## Round 0.1 · original submission · Major Revisions

I think the comments from Mary Christopher are especially helpful and constructive. I know you already made major changes to this article but hope you will be able to respond to the reviewer comments since both reviewers (and I) agreed that this was an interesting piece of work.

·

Basic reporting

1. The author lays out a good justification for the study, but it is long and should be more concise (suggest shorten by ~ a third).
2. Avoid repetition of the results, including table citations, in the Discussion.
3. The author refers often to "our" and "we", which does not correspond to a single-authored paper.

Experimental design

1. Introduction: I would argue that a journal's IF is "widely" rather than "sometimes even" extended to the evaluation of individual scientists (line 57).
2. Introduction: suggest avoid reference to the "Ingelfinger law", which detracts attention from the Matthew effect (too many names). I doubt most editors (or authors) realize or care whether there is a "law" prohibiting authors from duplicate publication; most simply view this as journal policy and the formal name doesn't aid understanding. On the other hand, the "Matthew effect" would benefit from more explanation; what exactly is it? what is "preferential attachment"? If appropriate, consider including this term in the hypothesis, since in the Discussion the main findings are interpreted as evidence of a Matthew effect.
3. Where you state (line 99) "However, this comparison was limited to pairs of publications." suggest indicate how/what comparing >2 publications will add to our understanding of the relationship between IF and citations.
4. The author indicates that "republished articles" were excluded, but it's not clear to me how these differ from simultaneously published articles. As I understand it, reporting guidelines have one original source and then are republished in multiple additional journals, often over an extended period of time.
5. The time frame of data extraction should be indicated.
6. It's not clear why two different citation databases were used (JCR/ISI and Scopus). Wouldn't it have been more accurate to use the same database of citations used to generate IFs? I also question the rationale to include and assign IF=0 to journals in Scopus that are not indexed by ISI; they are not necessarily at the bottom of the JCR category, they are simply not in the JCR category (e.g., they could be new journals that are not yet indexed).
7. Statistical analysis: Spearman's correlation is a nonparametric method for non-normally distributed data; log-transformation is another option for dealing with non-normally distributed data. Why were both methods used? Further, in Results the author variably refers to linear regression analysis, univariate linear regression, and multivariate analysis, none of which are explicitly mentioned in Methods (don't expect readers to know how Pearson's correlation coefficients are obtained). Importantly, the use of multiple nonparametric approaches gives the impression (even if not true) that the author did not consider what would be the best or most appropriate statistical test to use, but rather, tried a variety of correlation analyses to get the best results. The author should justify the use of more than one method for analyzing these data or choose a single method for correlation analysis.
8. The software used to analyze the data should be specified.
9. Specify whether you analyzed data for each reporting guideline separately, combined, or both.
10. Suggest reference 'Bradford distribution', which may not be familiar to readers; also, indicate the statistical test and parameters used to test data distribution.
11. Some Results (paragraphs 1 and 2) need to first be explained in Methods: (a) exclusion (and inclusion) criteria for the reporting guidelines (dates of publication, republications) and (b) how you defined simultaneous publication (specify and clarify in Methods)
12. Given the marked disciplinary differences in IF, it would be interesting to know the actual journals and disciplines involved, which could inform interpretation of the results. If possible, suggest include this information.

Validity of the findings

1. Statistical reporting: It's awkward to explain (and tabulate) that each logarithm unit of journal IF predicts a certain median change of logarithm units; this is technically correct, but in my view, graphs would be a more appropriate way to convey these results, especially since CIs are often wide. Also, when data are log-transformed, the results are then usually (albeit not always) transformed back to the standard unit of reference for interpretation (e.g., that each doubling of IF resulted in 4X more citations).
2. Given the relatively low number of data points for regression analyses, I would like to see graphs to better understand correlation and slope. Table 2 is complicated and makes it difficult to see important findings.
3. Interpretation of the results as 'strongly' correlated is appropriate in the Discussion but not in Results (unless you specifically define the correlation coefficient that qualifies as 'strongly' in Methods).
4. Table 1: how was an "Overall" year of publication of 2010 determined?
5. Suggest delete Figure 1: it's difficult to make sense of this. If the combined data were statistically analyzed, results can be reported in the text.
6. Discussion line 237 – consider expanding. IF is also considered an indicator of prestige and audience size, not just scientific value. Further, high-IF journals often have formidable marketing departments that drive media hype about articles and presumably citations.
7. In the first paragraph of Limitations, can the author comment on how the results might have been affected by these factors?
8. I would disagree with the characterization of a "convenience sample" (line 255); could the author explain?

Additional comments

(Summary) This study exploits the natural duplicate publication status of reporting guidelines to evaluate the correlation between citation rates and journal impact factor, extending previous work. I love the underlying premise: that IF itself contributes to higher citation rates, a reverse of the obvious, and a fresh and logical approach to thinking about citations. Although reporting guidelines are a unique subset of articles that come with their own unique citation practices, their publication in multiple journals with widely varying IFs is an appropriate and novel design for testing the hypothesis. The weakness of the study is its statistical analysis (which would benefit from refinement) and data presentation. With these improvements and a few clarifications in experimental design, the study provides an interesting and valid look at the 'autocorrelation' of IFs and citation rates.

Reviewer 2 ·

Basic reporting

The manuscript is quite clear and well-written. I've edited the PDF where I've observed minor grammar and style issues.

Most important, to adhere to Peer J's data policy, the author should provide the raw data, specifically the names of all journals, the usage statistics, parameters for each journal's definition of usage (for both PDF and HTML files), and any download data available from other repositories like PubMedCentral, if available. It's difficult to compare usage when one doesn't have the downloads.

The manuscript would benefit from a more thorough treatment of the relevant literature relating to biomedical publishing, including:

Lozano, Larivière, and Gingras (2012) http://onlinelibrary.wiley.com/doi/10.1002/asi.22731/abstract [which I believe introduces data that refutes your conclusions]

Are

Additionally, the author should list all procotols for data gathering.

Experimental design

As it stands, the conclusions would be impossible to reproduce without the methods and the journal names, as well as other details such as collection date, definitions of usage, and the raw data itself.

I think the research question was both relevant and meaningful.

'The Journal Citation Reports impact factors (IFs) are widely used to rank and evaluate journals, standing as a proxy for the relative importance of a journal within its field. However, numerous criticisms have been made of use of an IF to evaluate importance. This problem is exacerbated when the use of IFs is extended to evaluate not only the journals, but the papers therein. The purpose of this study was therefore to investigate the relationship between the number of citations and journal IF for identical articles published simultaneously in multiple journals.'

Nonetheless, I think the question can be more clearly stated to reflect the narrow scope of this study. While some of the limitations were included, that these were not research articles but rather reporting standards may in fact affect the results, and at the least, it would be more accurate for the author to reword the language in the manuscript (including the conclusions) to be precisely reflect the results for this particular study. The methods did not suffice for the conclusions to be generalized across other fields and other research article types.

Validity of the findings

Note: Some comments here may also dovetail or have been relevant in one of the other sections. See data availability requests in previous comments.

I am not a statistician, and while I could not identify any obvious shortcomings with Table 2 and (I believe that's Figure 1 - the scatterplots), I'd recommend a statistical review by the editor or by an additional reviewer with the relevant expertise.

As stated earlier in this review, I think the manuscript could benefit from a narrower statement of the research question, along with similarly focused conclusions (in particular the last sentence, below). It's not obvious to me that this: is fully supported by this study alone.

>>The impact factor of the journal in which an article was published was shown to influence the number of citations that article will gather over time. Similarly, the number of article downloads also influenced the number of citations, although to a lesser extent than the impact factor. This demonstrates that citation counts are not purely a reflection of scientific merit and the impact factor is, in fact, auto-correlated.

Additionally, other factors (which were not necessarily controller for) should be identified and discussed, and, if possible, included not just in the limitations section, but in the body of the paper such that these factors may be evaluated as to the extent of their influence on the validity of the findings.

-summary of number of or reach of promotional activities undertaken by the standards body (to promote the statement) e.g. http://www.eurekalert.org/pub_releases/2011-03/plos-nrg031011.php - what influences may these efforts have had on visibility? downloads? citations?
-do you have data identifying how users were alerted to each reporting statement in each journal? while this is unlikely, it might be interesting to know and to discuss the URLs user's visited before downloading the PDFs. What percentage came directly from the EQUATOR site? were readers first notified via an email blast, PubMed or Scopus Alert, notices at in-person meetings, etc.? in what context did usage (PDF downloads in this case) occur?
-while the reporting statements i believe were all free-to-read or open access (L247), what were the access control models of each of the journals? were there any effects of open access on downloads & accesses as compared to subscription-based journals? for STREGA, list the access models of the Lancet v. PLoS Medicine, for example, and check for correlations
-is there a reason why you evaluated only PDF downloads and not HTML accesses? it would seem that the latter is relevant too.
-in the limitations (L243-248), what is meant by the following? Can you be more specific in the body of your manuscript about these specific errors in the article counters, lost data, or data not collected? It's most useful to have details on the extent of this as a mitigating factor.
"There are a limited number of data points included in the correlation calculations, particularly
245 for analysis of article downloads. This was due to errors in the article counters, data being lost during
246 system migrations or the publishers’ not collecting that data. There is also the confounding issue of
247 publishers collecting article download data in different ways, and the rapid increase in e-journal use
248 between 2001 and 2006 (Schloegl & Gorraiz, 2010).

~The Standards for Reporting of Diagnostic Accuracy (STARD) were also published in Dutch. Does the author address different languages as a potential influence?

~ re: STARD; RSNA Radiology – 15 journals Some had been published early online. Were accesses also counted for EO?

~ re: STARD, Clinical Chemistry also published the background document - any relationship between that and downloads or cites of the STARD standard?

~GRIP [should be GRIPS] Genetic RIsk Prediction Studies

~PRISMA 2009 (7 journals) - the below may be worth looking at, again in terms of comparing the means of promoting or increasing visibility (here, in IFA)
"In order to encourage dissemination of the PRISMA statement, this article is freely accessible on bmj.com and will also be published in PLoS Medicine, Annals of Internal Medicine, Journal of Clinical Epidemiology, and Open Medicine.
Recent surveys of leading medical journals evaluated the extent to which the PRISMA Statement has been incorporated into their Instructions to Authors. In a sample of 146 journals publishing systematic reviews, the PRISMA Statement was referred to in the instructions to authors for 27% of journals; more often in general and internal medicine journals (50%) than in specialty medicine journals (25%).[13] These results showed that the uptake of PRISMA guidelines by journals is still inadequate although there has been some improvement over time."

Additional comments

In the manuscript 'Auto-correlation of journal impact factor for consensus research reporting statements: a cohort study,' Daniel R. Shanahan conducts an analysis that has the potential to offer a meaningful way to assess the influence of Journal Impact Factor (JIF or IF) on citations and downloads for identical material.

This in fact has been a longstanding thought experiment for many, and it is, therefore, refreshing to see the study actually performed.

While the piece is interesting and has potential merit, to confirm the validity of the findings. several modifications are suggested in this review, in particular including in the manuscript body or supplemental files (exact placement at the editor's discretion) additional identifying information and all raw data as well as detailed methods for data gathering.

Regardless, I recommend toning down the conclusions to be specific to this type of material (not a research investigation, but rather, a consensus reporting statement) and this type of journal.

Annotated reviews are not available for download in order to protect the identity of reviewers who chose to remain anonymous.

---

## Round 0.2 · accepted · Accept

Thanks very much for your detailed responses to the reviewer comments -- I'm glad you found them helpful.